# Overweight, Obesity, and Depression in Multimorbid Older Adults: Prevalence, Diagnostic Agreement, and Associated Factors in Primary Care—Results from a Multicenter Observational Study

**DOI:** 10.3390/nu17081394

**Published:** 2025-04-21

**Authors:** Daniel Christopher Bludau, Alexander Pabst, Franziska Bleck, Siegfried Weyerer, Wolfgang Maier, Jochen Gensichen, Karola Mergenthal, Horst Bickel, Angela Fuchs, Ingmar Schäfer, Hans-Helmut König, Birgitt Wiese, Gerhard Schön, Karl Wegscheider, Martin Scherer, Steffi G. Riedel-Heller, Margrit Löbner

**Affiliations:** 1Institute for Social Medicine, Occupational Health and Public Health, University of Leipzig, 04103 Leipzig, Germany; 2Central Institute of Mental Health, Medical Faculty Mannheim, Heidelberg University, 68159 Mannheim, Germany; 3Department of Neurodegenerative Diseases and Geriatric Psychiatry, University Hospital Bonn, 53127 Bonn, Germany; 4Institute of General Practice and Family Medicine, University Hospital of LMU Munich, 80336 Munich, Germany; 5Institute of General Practice, Goethe-University Frankfurt, 60590 Frankfurt am Main, Germany; 6Department of Psychiatry, Technical University of Munich, 81675 Munich, Germany; 7Institute of General Practice, Medical Faculty of the Heinrich Heine University Düsseldorf, 40225 Düsseldorf, Germany; 8Department of Primary Medical Care, Center of Psychosocial Medicine, University Medical Center Hamburg-Eppendorf, 20246 Hamburg, Germany; 9Department of Health Economics and Health Services Research, Hamburg Center for Health Economics, University Medical Center Hamburg-Eppendorf, 20246 Hamburg, Germany; 10MHH Information Technology, Hannover Medical School, 30625 Hannover, Germany; 11Department of Medical Biometry and Epidemiology, Center for Experimental Medicine, University Medical Center Hamburg-Eppendorf, 20251 Hamburg, Germany

**Keywords:** body mass index, depression, Geriatric Depression Scale, late life, multimorbidity, obesity, primary care

## Abstract

**Background/Objectives**: Obesity and depression, in conjunction with multimorbidity, are interconnected conditions increasingly managed in general practitioner (GP) settings, yet these associations remain insufficiently studied in older patients. This study investigates the prevalence of depression across different body mass index (BMI) classes and includes age and gender differences in multimorbid older patients, offering a novel perspective on subgroup-specific patterns. Further the agreement between GP depression diagnoses and the Geriatric Depression Scale (GDS) is studied and patient-specific factors that may affect the agreement are explored, aiming to improve future diagnostics for vulnerable subgroups. **Methods**: Data were provided by the baseline assessment of the MultiCare Study, a prospective multicenter observational cohort of multimorbid patients aged 65+ years recruited from 158 GP practices across eight study centers in Germany. Data from 2568 study participants were analyzed based on GP-coded International Classification of Diseases (ICD) diagnoses, structured GP questionnaires, and patient questionnaires. Assessments included data on the BMI and depression (15 item version of the GDS). Agreement between GP diagnoses of depression and GDS assessment was measured using Cohen’s kappa. Four logistic regression models were used to examine the effects of patient-specific factors on the agreement of depression diagnosis (match or mismatch). **Results**: GPs diagnosed depression in 17.3% of cases, compared to the detection of depressive symptoms in 12.4% of the patients by GDS (cut-off ≥ 6 points). The highest prevalence rates were observed in patients with obesity class III (25.0% by GP; 21.7% by GDS). Women were significantly more likely to receive a depression diagnosis by a GP across most BMI classes (except obesity classes II and III). The detection of depressive symptoms by GDS was significantly more prevalent in older multimorbid obese patients (≥75 years), except for patients with obesity class III. The overall agreement between GP diagnosis and GDS assessment was weak (κ = 0.156, *p* < 0.001). The highest agreement was found for people with obesity class III (κ = 0.256, *p* < 0.05). Factors associated with a True Positive depression diagnosis (match by both GDS and GP) were female gender (odds ratio (OR) = 1.83, *p* < 0.05), widowhood (OR = 2.43, *p* < 0.01), limited daily living skills (OR = 3.14, *p* < 0.001), and a higher level of education (OR = 2.48, *p* < 0.01). A significantly lower likelihood of a False Negative depression diagnosis was found for patients with obesity class III. **Conclusions**: This study highlights the significant prevalence of depression among multimorbid older adults across different BMI classes, particularly in those with obesity class III. The weak diagnostic agreement between GP diagnosis and GDS assessment suggests a need for improved diagnostic practices in primary care. Implementing standardized screening tools and fostering collaboration with mental health specialists could enhance the identification and management of depression in this vulnerable population.

## 1. Introduction

Recognizing their growing global impact, the World Health Organization (WHO) has increasingly been monitoring both obesity and depression, spurring research to investigate effective strategies for prevention, diagnosis, and integrated care and underscoring the need to address these pressing global health issues [1,2,3]. Obesity is a common non-communicable disease with far-reaching health consequences. It has shown a rapid increase in global prevalence over the last decades, with adult obesity rates more than doubling since 1990 [4,5]. The problem of obesity cuts across generations; among older adults, the rate of overweight and obese individuals has been rising, with particularly large increases in the last 40 years [6,7]. Overweight and obesity, which are quantifiable by body mass index (BMI), increase with age in both sexes and cause a range of adverse health outcomes [8]. Examples include the development of a metabolic syndrome consisting of type 2 diabetes or cardiovascular manifestations, as well as other conditions like steatohepatitis, osteoarthritis, the increased likelihood of cancer, and the development of depressive disorders [9]. Often, more than one of these chronic diseases coexist, creating multimorbidity. Obesity and multimorbidity are closely associated [10]. As overweight and obesity levels increase, so does the prevalence of multimorbidity [11]. This association has become more evident over the past 25 years, with increasing BMI leading to a greater prevalence of multimorbidity [12]. This association between obesity and multimorbidity is particularly strong in older patients [12].

In old age, in particular, lifestyle factors appear to play a significant role in modifying BMI, with physical activity playing the most critical role [13]. Genetic; biological (age and gender); and social determinants of health (such as education, income, and having a partner) interact with these factors in a complex way, in part, because they influence eating and activity behavior [9].

Depression is a common, growing, and increasingly relevant condition, recently seeing a rise in public awareness and media coverage [14]. Worldwide, it is among the leading contributors to disability-adjusted life years (DALYs) across a broad range of age groups. The trend over the last 30 years has been upward, with a more than 60% increase in DALYs due to depressive disorders [15]. For younger and middle-aged adults, depression ranks between first and third as the most common cause of years lived with disability (YLDs); for older adults (aged 65 or older), depression is still one of the top five causes for years spent with a disability. The trend of depression increasingly becoming a cause of disability has been particularly pronounced in recent decades, especially among older patients and compared to other diseases [16].

For older patients, depression is influenced by the number of additional medical conditions, and multimorbidity is related to both the presence and the severity of depression [17,18]. Depressive symptoms are significantly more frequent in multimorbid patients, and furthermore, multimorbid patients also show higher depression severity levels than non-multimorbid persons [18]. A cumulative effect of the number of conditions on the prevalence of depression is assumed [19].

As with the multimorbidity that often accompanies it, there is strong evidence that obesity has a major impact on mental health and vice versa [20]. Especially for younger patient groups, it has been shown that there is a positive association between childhood or adolescent obesity and depression. Similarly, more severe depressive symptoms were found in obese adolescents [21]. The question of an association in old age has been studied less frequently, but associations between depression and obesity in older populations have also been pointed out. For instance, studies indicate a bidirectional relationship between obesity and depression [22], with the likelihood of depression appearing to increase with increasing the BMI [23].

The relationship between differentiated BMI classes and depression in a multimorbid age cohort has so far been little researched. Furthermore, it has not yet been investigated whether biological and social determinants of health are adequately taken into consideration when diagnosing depression in general practitioner (GP) practices, where, in addition to multimorbidity management, more than half of the ongoing psychiatric care (estimated 60%) and the majority of antidepressant prescriptions are carried out [24]. Finally, previous studies point out inconsistencies in the diagnosis of depression between GPs and screening instruments such as the Geriatric Depression Scale (GDS) [25], as well as gender differences in this regard [26].

This study aims to investigate possible gaps in the outpatient primary care treatment of depression in obese, multimorbid patients in old age and to identify determinants of health influencing the diagnosis. It is the first study to explore age- and gender-specific depression prevalence across BMI classes and examine diagnostic agreement between GP diagnoses and GDS assessments while accounting for the BMI in a large sample of multimorbid older adults in primary care.

The following research questions will be addressed:What are the age- and gender-specific prevalence rates of depression, and how do they differ between individuals with under- and normal weight, overweight, and with varying degrees of obesity?What is the rate of agreement comparing depression diagnoses made by GPs and detected by a validated instrument (GDS) with regards to different BMI classes?Which factors are associated with a match (True Positive or True Negative), and which factors are associated with a mismatch (False Positive or False Negative) between the GP’s clinical judgment and the GDS-based screening outcome? What role does the BMI class play in this context?

## 2. Materials and Methods

### 2.1. Sample

Data were obtained from the baseline assessment of the MultiCare study, a prospective multicenter observational cohort in Germany. The study includes randomly selected GP patients aged 65 years or older with multimorbidity and comprises a baseline survey and three follow-up surveys. The baseline survey took place between July 2008 and October 2009. The mean age of the patient collective at baseline was 74 years. A total of N = 3189 patients were enrolled, in accordance with the study protocol, which targeted a minimum enrollment of 3050 patients [27]. The proportion of women was 59%, and 56% of the study population were married [28]. This sample can be considered representative of the population distribution of Germany in terms of gender and marital status for the age group of 65 years and older [29].

From 158 GP practices in the eight participating German study centers in Bonn, Düsseldorf, Frankfurt/Main, Hamburg, Jena, Leipzig, Mannheim, and Munich, study participants in the relevant age group with a date of birth between July 1923 and June 1943 who had consulted their GP during the last quarter were identified. Using a random number table, 50 patients per GP were selected from all eligible patients, contacted by the GP, and invited to participate in the study. Interested patients were then given information about the study and asked for written informed consent.

The study excluded individuals who were residing in nursing homes (short life expectancy) or who had a severe illness expected to be fatal within three months according to the GP. Further exclusion criteria included insufficient proficiency in speaking and reading German, inadequate capacity to provide consent (e.g., dementia), inability to participate in interviews (e.g., blindness and deafness), being unfamiliar to the GP (e.g., accidental consultation), or current participation in other studies.

Only patients who simultaneously had at least three chronic diseases from a defined (International Classification of Diseases [ICD]-10 codes) catalogue of 29 diseases were eligible for inclusion in the study. Following this selection process, a total of 3317 patients agreed to participate in the study. As part of the data analyses, 128 patients had to be excluded retrospectively due to situational circumstances (e.g., premature death) and 621 patients due to incomplete or missing data on the factors investigated. The final analytical sample comprised 2568 patients, as illustrated in Figure 1.

### 2.2. Data Collection and Assessment Procedure

The multimorbid disease burden of the patients was recorded using three approaches: Firstly, a chart review was used to record all ICD-10 diagnoses for each patient as recorded in the GP’s database in the last quarter. Secondly, structured GP interviews were conducted. Thirdly, each patient was interviewed by the same interviewer over the entire duration of the study using standardized clinical questionnaires.

Patient age, gender, and current diagnoses were obtained through a chart review of the respective GP’s database. Medical checkup data such as the patient’s BMI were transmitted by the GP. Sociodemographic data (education level determined by CASMIN classification, marital status, and income) were collected from the direct patient interview. Furthermore, information on the mental state of the patients was obtained in direct dialogue with the patients using the GDS. The ability to carry out daily activities independently was assessed using the Instrumental Activities of Daily Living (IADL) Scale and pain perception was quantified using the Graded Chronic Pain Scale (GCPS). In the interviews, patients also provided information on physical activity (International Physical Activities Questionnaire IPAQ-S7S) and social integration (F-SozU K14).

### 2.3. Sociodemographic Data

The sociodemographic variables collected were age, gender, education level, marital status, and income. Two age groups were distinguished (young old people: 65–74 years and older old people: over 75 years) [30]; the education level was determined using the CASMIN classification (low, medium, and high); and marital status was categorized into married, single, divorced, and widowed. The monthly household net income was recorded and assigned to 1 of 23 graduated income levels, ranging from 1 (below 150 EUR) to 23 (10,000–18,000 EUR). These income levels were converted into continuous values (e.g., 150–300 EUR corresponds to 225 EUR) to calculate the equivalized disposable income. For international comparability, the equivalized disposable income according to the modified OECD equivalence scale was used, whereby the head of household was weighted with a factor of 1 and each additional household member with a factor of 0.5 (≥15 years) or 0.3 (<15 years), according to age [31]. This equivalized disposable income was then scaled for a better overview and divided by a factor of 1000.

### 2.4. Resources and Risk Factors

Body weight and height were measured, and the BMI was calculated (body weight in kilograms divided by squared height in meters). According to the guidelines, the classification was made into BMI classes (under- and normal weight ≤ 24.9 kg/m^2^, overweight 25–29.9 kg/m^2^, and obesity ≥ 30 kg/m^2^) [32], whereby obesity was additionally subdivided into its obesity classes I (30–34.9 kg/m^2^), II (35–39.9 kg/m^2^), and III (≥40 kg/m^2^).

The IADL score was used to assess whether participants could still manage an independent daily life. The skills assessed include aspects of household management, personal health care, telecommunication skills, financial and schedule management, and mobility [33]. The summary score usually varies from 0 to 8 for women and, to prevent gender bias, from 0 to 5 for men. Lower scores signify reduced function and dependence on assistance, while higher scores indicate enhanced function and independence [34]. For better comparability of the obtained score between men and women in this cohort, the three items “food preparation”, “housekeeping”, and “laundry” were not included in the analysis, so that each participant could now obtain a maximum score of 5. For better interpretation of the score, dichotomization into not limited (5 points) versus limited (0–4 points) was performed.

To determine the severity of chronic pain, the Graded Chronic Pain Scale (GCPS) was used [35]. Based on the first three questions concerning pain intensity (involving the items “Pain Right Now”, “Worst Pain” during the last 4 weeks, and “Average Pain” during the last 4 weeks) the “Characteristic Pain Intensity” was calculated, providing information on the presence of pain (yes versus no) and the subjective intensity of perceived pain on a scale from 0 to 100.

Through a separate questionnaire listing the chronic diseases included in the study, multimorbidity was quantified by adding up the chronic diseases as reported by the patients [27].

Physical activity was assessed using the International Physical Activities Questionnaire (IPAQ-S7S). The self-reported physical activity of the last 7 days was assessed by a series of 7 items. The activity level was then classified as a low, moderate, or high activity level. The evaluation of the data is based on the “Guidelines for Data Processing and Analysis of the IPAQ” [36].

The questionnaire F-SozU in its short form (K-14) was used to survey social integration by measuring perceived and anticipated social support with 14 items using a five-point Likert scale. The participants were presented with 14 positively formulated statements about their social environment, and the extent of their agreement was assessed on a scale ranging from 1 (strongly disagree) to 5 (strongly agree). The individual item values were summed and divided by the number of items to yield an average, with a score of 5 corresponding to the maximum and a score of 1 corresponding to the minimum perceived social support. To interpret the obtained value, one can refer to normalization against a representative German sample, where an average scale value of 3.97 with a standard deviation of 0.68 serves as a reference [37,38].

### 2.5. Dependent Variables

GDS assessment. The GDS was used in its abbreviated form (GDS-15) in the patient questionnaire as an established assessment instrument for the detection of depression in older age [39]. In the questionnaire, patients can respond affirmatively or negatively to 15 questions regarding mental well-being (e.g., “Have you dropped many of your activities and interests?”). A total score is then calculated based on the responses. A total of 0–5 points was considered normal, while a cut-off of ≥6 points defined the assumption of a depressive disorder [40,41].

GP-diagnosis. In order to determine whether a diagnosis of depression had been detected by a doctor, the responsible GP was asked in a standardized questionnaire whether the patient had a depression diagnosis according to ICD groups F32-F33 of the ICD-10 medical classification list [27,28,42].

The abovementioned assessments of depression were used to describe either a concordance or discordance between the GP’s clinical judgment and the GDS-based screening outcome, hereafter referred to as “diagnostic agreement” in this study. For clarity, the following categories were defined:

True Positive (Indication of depression by both the GDS and the GP);

True Negative (No indication of depression by either the GDS or the GP);

False Positive (No indication of depression by the GDS but clinical diagnosis of depression by the GP); 

False Negative (Indication of depression by the GDS but no clinical diagnosis of depression by the GP).

### 2.6. Statistical Analyses

All statistical analyses were performed using STATA SE, version 18 (StataCorp., College Station, TX, USA) for Windows. The significance level for the analyses was set at alpha ≤ 0.05. Sociodemographic and clinical characteristics of the study sample were presented descriptively, and gender differences were analyzed statistically (a chi-squared test was used for categorical variables, a *t*-test for normally distributed continuous variables, and a Wilcoxon rank-sum (Mann–Whitney) test for non-normally distributed continuous variables).

A descriptive presentation of the prevalence of depression (GDS assessment and GP diagnosis respectively) was made for the entire population, as well as categorized into BMI classes (under- and normal weight, overweight, obese, and stratified into obesity classes I–III). In the same course, the prevalence of depression was additionally presented categorized according to gender and according to age group (younger old people 65–74 years; older old people ≥ 75 years) and each tested for significant group differences with a chi-squared test or a Fisher’s exact test (in case of small sample sizes with observations of less than 5).

The depression assessments according to the GDS and GP questionnaire were then compared for agreement in a second analysis, using Cohen’s kappa as a statistical measure for interrater reliability. Bias-corrected Cohen’s kappa with a 95% confidence interval was calculated for the total population, as well as separately for the categories of the respective BMI classes.

In order to examine the patient-specific factors gender, age, marital status, education level, income, BMI, independence in daily life, perceived pain, number of chronic diseases, physical activity, and social support with regards to their influence on the indication of depression according to the GP’s clinical judgment and the GDS-based screening outcome, four logistic regression models were calculated to estimate the probability of a concordance (True Positive or True Negative) or a discordance (False Positive or False Negative) in indicating depression, respectively, depending on the presence of patient-specific risk and protective factors. The calculated odds ratios (ORs) were tested for significance and described the association between the patient-specific factor and the corresponding scenarios of either a depression diagnosis or no depression diagnosis made by the GP and their (mis)match in this regard with the assessment by the GDS.

## 3. Results

### 3.1. Sample Characteristics

The study population of 2568 participants consisted of people aged 65–86 years, 1498 (58.3%) of whom were female (see Table 1). The majority of both genders were married, while noticeably the majority of the widowed were female.

While only one-fifth (21.9%) of the men were of under- and normal weight, this was the case for around one-fourth (26.2%) of the women. The largest group in both sexes was overweight (50.6% of men and 38.7% of women). Obese patients made up the second largest group in terms of BMI class (27.6% of men and 35.1% of women), with class I obesity being the most common form in both genders.

### 3.2. Prevalence of Depressive Symptoms According to Body Mass Index, Gender, and Age

Using the GDS, a total of 319 patients (12.4%) were classified as depressed, while GPs diagnosed 444 patients (17.3%) with depression (see Table 2). In general, across almost all BMI classes (with the exception of obesity class II), GPs more frequently issued a diagnosis of depression compared to the GDS (under- and normal weight: 18.5% by GP versus 12.3% by GDS, overweight: 16.4% by GP versus 10.4% by GDS, and obesity in total: 17.5% by GP versus 15.4% by GDS).

In both diagnostic approaches, with the slight exception of obesity II in GP-diagnosed depression, more severe degrees of obesity also showed increased depression prevalence in their respective BMI class (obesity I: 17.1% by GP and 14.0% by GDS and obesity II: 16.5% by GP and 18.4% by GDS, with the maximum observed for obesity III: 25.0% by GP and 21.7% by GDS).

A gender-specific analysis showed that GPs diagnosed depression significantly more often in women. This phenomenon was found for the overall study sample (*p* ≤ 0.001) and for women with under- and normal weight (*p* = 0.009), overweight (*p* ≤ 0.001), and obesity in total (*p* ≤ 0.001), as well as for obesity class I (*p* ≤ 0.001). In contrast, when assessed by GDS, only women as a group across all BMI classes of the overall study sample (*p* ≤ 0.001) and overweight women especially (*p* = 0.006) showed a significantly higher rate of depression. All other BMI classes did not show any significant differences in terms of gender distribution when assessed by GDS.

When looking at age-specific differences, only amongst the under- and normal weight did the GP-made depression diagnoses show a significant shift (*p* = 0.012) towards patients aged below 75 years. Meanwhile, the GDS revealed a contrary shift, with significantly increased depression rates on the part of older patients aged 75 years or above for the overall study sample (*p* = 0.040) and for obesity in total (*p* = 0.039), as well as obesity I (*p* = 0.030) and obesity II (*p* = 0.036) in particular. Only for obesity III did the GDS show a significant depression prevalence shift towards the younger patients aged below 75 years (*p* = 0.043).

### 3.3. Diagnostic Agreement of Depression Analyzed by BMI Classes

When comparing the agreement (Figure 2) between GDS assessment and GP diagnosis, a weak agreement was found for the overall study sample with a kappa coefficient [95% confidence interval, *p*-value] of κ = 0.156 [95% CI: 0.133–0.174, *p* < 0.001] (illustrated in Figure 3). Likewise, weak agreements were found when broken down by BMI class, with κ = 0.179 [95% CI: 0.120–0.231, *p* < 0.001] for under- and normal weight, κ = 0.114 [95% CI: 0.093–0.162, *p* < 0.001] for overweight, and κ = 0.185 [95% CI: 0.157–0.241, *p* < 0.001] for obese patients in total. A further distinction into obesity classes showed similarly weak agreements with κ = 0.194 [95% CI: 0.148–0.270, *p* < 0.001] for obesity I and a non-significant κ = 0.109 [95% CI: −0.016–0.184, *p* = 0.088] for obesity II but a slight improvement of κ = 0.256 [95% CI: 0.090–0.510, *p* < 0.05] for obesity III. Similarly, Receiver Operating Characteristic (ROC) analyses indicate that the GDS does not adequately fit with the GP diagnosis in distinguishing between individuals with and without depression symptoms (area under the curve = 0.640 [95% CI: 0.623–0.656]; standard error = 0.0134; sensitivity at cut-off score 6 = 0.43; specificity at cut-off score 6 = 0.78) (see Appendix A).

### 3.4. Patient-Specific Factors and Their Influence on the Outcome of GP Diagnosis Versus GDS-Based Depression Assessment

Four logistic regression models were used to examine the influence of patient-specific factors on the diagnostic agreement of indicating depression (see Table 3). True Positive describes the indication of depression by both the GDS and the GP, while True Negative shows no indication of depression by either the GDS or the GP. False Positive describes no indication of depression by the GDS but clinical diagnosis of depression by the GP, and False Negative describes the indication of depression by the GDS but no clinical diagnosis of depression by the GP.

### 3.5. Concordance in Indicating and Not Indicating Depression

Patients who were significantly more likely to suffer from depression (identified through a match by both GDS and GP) in the True Positive model were women (OR = 1.83 [95% CI: 1.05–3.18]), widowed (OR = 2.43 [95% CI: 1.47–4.01]), those with limited IADL (OR = 3.14 [95% CI: 1.98–5.00]), and people with a high level of education (OR = 2.48 [95% CI: 1.30–4.72]). Along with divorced individuals, those also showed a lower probability of receiving a matching non-indication of depression in the True Negative model (women: OR = 0.58 [95% CI: 0.47–0.73], widowed: OR = 0.75 [95% CI: 0.59–0.94], limited IADL: OR = 0.61 [95% CI: 0.49–0.76], and divorced: OR = 0.67 [95% CI: 0.47–0.95]).

In contrast, the factors associated with a significantly increased probability of concordant non-indication of depression were age of ≥75 years (OR = 1.34 [95% CI: 1.09–1.65]), higher income (OR = 1.38 [95% CI: 1.16–1.64]), high physical activity (OR = 1.68 [95% CI: 1.27–2.22]), and social support (OR = 1.74 [95% CI: 1.51–2.00]), the latter being simultaneously associated with a lower probability of a True Positive indication of depression (social support: OR = 0.36 [95% CI: 0.28–0.46]).

Regarding BMI classes, there was only one significant effect with a slightly increased probability of receiving a matching non-indication of depression belonging to the obesity I class (OR = 1.35 [95% CI: 1.02–1.78]). The other BMI classes did not show any significant effects regarding matching depression diagnoses.

### 3.6. Discordance in Indicating and Not Indicating Depression

Women were significantly more likely than men to receive False Positive depression diagnosis from their GP (OR = 2.01 [95% CI: 1.51–2.68]). There were no significant associations between any BMI class and a False Positive depression diagnosis.

While limited daily activities (OR = 2.14 [95% CI: 1.54–2.99]) and, to a lesser extent, pain intensity (OR = 1.02 [95% CI: 1.01–1.02]) and the number of chronic diseases (OR = 1.13 [95% CI: 1.07–1.18]) were associated with a significantly increased likelihood of a False Negative depression diagnosis, the factors income (OR = 0.47 [95% CI: 0.33–0.66]), high physical activity (OR = 0.37 [95% CI: 0.22–0.64]), and social support (OR = 0.49 [95% CI: 0.40–0.60]) were significantly less likely to be associated with False Negative depression diagnoses. Looking at the BMI classes, there was a significantly lower likelihood of a False Negative depression diagnosis exclusively for the obesity III class (OR = 0.33 [95% CI: 0.13–0.84]).

## 4. Discussion

The aim of this study was to investigate the prevalence of depression in different BMI classes among multimorbid older patients (aged 65+ years) and to explore differences amongst obesity grades I to III, as well as possible gender and age differences. We also examined potential differences in the identification of depression between the GP’s clinical judgment and the outcomes of the GDS, a scientifically well-established screening instrument. In addition to analyzing for agreement, physical and biopsychosocial factors were examined for their influence on depression diagnosis with the aim of identifying possible weaknesses in depression screening and sensitizing GPs, who are important providers of basic psychiatric care for this patient group.

It is interesting to note that the percentage of overweight people (according to BMI) reported in European and German studies of older age cohorts is very similar to our data, while the percentage of under- and normal weight patients in our multimorbid study sample is about 10–18% lower with a roughly equal increase in the percentage of obese people instead [43,44].

Like a previous study comparing depression diagnoses by GPs and GDS assessments, this study confirms that GPs diagnosed depression overall more frequently than the GDS [25]. The data from the present study show that this phenomenon remained constant across almost all BMI classes (except for obesity II). The lowest prevalence of depression was measured by GPs and GDS in the overweight class and differed by more than 10 percentage points from the highest prevalence of depression, which was found in the obese III class. Significant gender differences appeared to exist particularly when diagnosed by GPs, as women were significantly more likely to receive a depression diagnosis across almost all BMI classes. When the obese subgroups were broken down, this phenomenon was particularly significant in the early stage (obesity I). In contrast, significant age group differences dominated in the GDS. Except for the obesity III class, older patients (≥75 years) in the other obesity classes tended to suffer from depression more frequently than the younger comparison group (65–74 years).

The overall weak agreement between GP diagnosis and GDS assessment in direct comparison with a kappa of 0.16 was to be expected based on similar previous studies of this multimorbid cohort, which reported a κ = 0.18 [25] and a moderate κ = 0.28 in a different age cohort aged 75+ years [26]. These studies suggested this could be due to the two approaches taking different variables into account when diagnosing depression, with screening tools like the GDS being more sensitive to subliminal depression that does not qualify (yet) for a clinical diagnosis [26]. Even when controlling with other depression screening tools apart from the GDS, the kappa values remain within this range [26], so that the weak to moderate agreement appears to be a GDS-independent effect. When looking at BMI classes, the analysis did not show any better or worse agreement in individual subgroups. This goes against the expectation that weight or obesity stigma might affect GP diagnoses of mental disorders like depression, as suggested in a previous study [45]. In fact, the statistically highest agreement was observed for people with obesity III. Although the analysis of the kappa values indicates a poor agreement, it should be noted that these low kappa values still bring an agreement of around 75%, which means that, despite recognizable discrepancies, the GPs and the GDS came to the same conclusion in three quarters of the cases. For international comparison, this result can be juxtaposed with an Indian study with 306 participants (60+ years) in which there was, in principle, a good agreement between GDS and ICD-10 diagnosis with a kappa of 0.85, although the comparison was not based on the documented GP diagnosis of depression but on the ICD-10 criteria findings surveyed during the study, which could explain the higher agreement rate [46].

In our cohort, the diagnosis of depression was influenced by gender. GPs appear to be more sensitive to female patients and more likely to correctly diagnose existing depression as such. At the same time, women are also often mistakenly diagnosed with depression, whereas men are less likely to be suspected of having depression, and thus, more depression is correctly ruled out. The postulation of this effect is supported by research that attributed an increased probability of a GP diagnosis of depression to women [26].

GPs often correctly ruled out depression in patients over the age of 75. Widowhood, which incidentally mostly affects women, was usually correctly recognized as a risk factor and an indication of an accompanying depression. However, widowed and divorced people were probably also more often overdiagnosed by “cautious” doctors who less frequently correctly ruled out depression in them. The relationship between widowhood and depression was recently analyzed in a meta-analysis and found that the prevalence of depression, although decreasing over time from around 38% to 11%, remains above average, particularly in the first five years after the loss of a partner [47].

Higher education was associated with an increased likelihood of correct depression recognition. Similarly, people with higher incomes were more likely to correctly be excluded from depression, and the risk of overlooked depression was lower. These effects could possibly be due to better access to the healthcare system and preventative measures for these groups, as well as a better ability to articulate their own symptoms. Other studies confirm the protective effect of education and income on the prevalence of depression [48] and suggest that cytokines like IL-1β, TNFα, or the NF-κB pathway, which play a role in both obesity and depression, could explain the link [49]. Education as protective against obesity, which is more strongly correlated in women than in men, could therefore also have an influence on depression [50].

While physically active patients according to the IPAQ were more likely to be correctly ruled out for depression, people with limited daily activities according to the IADL showed diagnostic uncertainties with more correct depression diagnoses, fewer correct depression exclusions, and an increased probability of overlooked depression. Meanwhile, both pain and the number of chronic diseases showed only a weak influence on the diagnosis of depression.

In contrast, social support seemed to act as a protective factor against depression, leading more likely to correct depression exclusions. It was also associated with fewer overlooked depression diagnoses, potentially due to diagnosis-promoting hints from the patients’ social environment. Social support has been described in the literature elsewhere as a possible protective factor against depression, particularly in lonely older people [51,52]. On the other hand, socially supported patients in our study were also less likely to receive correct positive depression diagnoses, which could indicate that depression may be less obvious and therefore more difficult to recognize in socially well-integrated patients.

Contrary to expectations, the BMI class did not play a pronounced role in the process of diagnosing and the diagnosis quality of depression. Significance was only found in minimum and maximum levels of obesity. Mild obesity was associated with an increased probability of correctly ruling out depression. This may be due to doctors being sufficiently sensitized to recognize physical manifestations related to obesity that can mimic depressive symptoms and therefore not immediately assuming a diagnosis of depression. On the other hand, extreme obesity seemed to reduce the likelihood of existing depression being overlooked by the GP. It is possible that most GPs are aware of the interplay between (extreme) obesity and depression. The literature also provides the explanation that, in particular, multimorbid obese people, in view of low self-esteem and learned helplessness, are more likely to utilize health services and seek help [53]. Overall, we found only a few significant correlations, which were not systematic and tended to relate to marginal categories. This leads to the conclusion that, in the German outpatient GP setting, contrary to original concerns, depression is probably diagnosed largely independently of body weight; thus weight-associated stigma appears to play less of a role than previously assumed [54,55]. At the same time, a detailed analysis of the differentiated degrees of obesity revealed a possible increased awareness of GPs for mental health in patients with extreme obesity (obesity III). Nevertheless, the results could also indicate that weight-associated mental consequences are not (yet) sufficiently reflected, as there was no systematic positive influence on correct diagnoses (matches). Either way, there does not appear to be a systematically increased rate of overlooked depression in overweight or obese patients.

### Strengths and Limitations

The strengths of this study were its multicenter design and a large sample size of a so far insufficiently investigated patient collective of multimorbid older patients, a group that has growing relevance in the future due to demographic changes.

Limitations of the present study include the fact that only cross-sectional data from the baseline survey between July 2008 and October 2009 were analyzed, which means that correlations could be identified but no causal link could be proven. This study design cannot determine the causal direction of the bidirectional relationship between obesity and depression [22,23]. Future studies should examine these associations using more recent data and longitudinal study designs.

The GDS with 15 items was used as a well-established screening standard in our patient group (using a cut-off of ≥6 points for a likely depressive disorder), although more sensitive or restrictive cut-off values of ≥5 to ≥8 points are also discussed in the literature, which may, however, be associated with the risk of over- or underestimating the prevalence of depression [56]. It can be assumed that using a different cut-off value in our study would have influenced the estimated prevalence rates and the level of agreement between GDS assessments and GP diagnoses. A recent meta-analysis identified a cut-off of ≥8 as providing the best agreement with the Structured Clinical Interview for Diagnostic and Statistical Manual of Mental Disorders (SCID), the diagnostic gold standard. However, due to substantial heterogeneity across studies, this threshold was not considered suitable for routine clinical use [56].

It should be noted that the GDS remains a screening tool and can only provide indications of depression but cannot be considered the gold standard for its diagnosis. Like other screening tools, the GDS has inherent limitations due to its respective sensitivity and specificity, which influence its positive predictive value and utility in accurately identifying true cases of depression [57]. The diagnosis by the GP was assumed in our study exclusively on the basis of coded ICD diagnoses, although there are indications that this approach underestimates the “awareness” of GPs and therefore rather represents a “lack of registration” instead, which could be ameliorated by considering further indications from the medical record such as free text notes or antidepressant medication prescriptions [58]. The experience of GPs in the treatment of depressive disorders is another factor that seems to influence the diagnosis [26] and was not considered further in the present study.

In addition, other factors that we did not investigate may have influenced our findings. Previous studies highlight lifestyle-related variables such as diet, alcohol use, and smoking as relevant variables that can affect the relationship between obesity and depression. [59]. While the GDS is designed to minimize the impact of somatic health on depression screening in older adults, prior research suggests that the clinical diagnosis of depression, as made by GPs, may be confounded by residual symptoms, such as fatigue or weight loss in the context of cancer diseases [60]. Although these factors were not considered in our study, they might have contributed to differences in diagnostic agreement. These factors should be taken into account in future research.

The four logistic regressions in our study were able to reveal some clear correlations between the factors examined and the diagnostic outcome; however, the pseudo-R^2^ values of our models with the factors examined only explain up to a maximum of just about 20% of the variance and varied within the same dataset in our four different models from 0.035 to 0.198. More research is needed in the future to identify possible other factors that could play a role in this complex interaction.

## 5. Conclusions

The present study is the first to examine age- and gender-specific prevalence rates of depression across different BMI classes in a large sample of multimorbid older adults in primary care. It is also the first to provide information on diagnostic agreement between GP diagnoses and GDS assessments while taking the BMI into account. The study highlights important factors associated with a poorer concordance between GP’s clinical judgment and the GDS-based screening outcome among older adults with multiple health conditions in primary care. The results provide important insights into potential target groups that could benefit from intervention programs designed to improve depression diagnostics and treatment. A substantial proportion of multimorbid patients suffer from depression across all BMI classes, with GP patients in obesity class III identified as having the highest risk of depressive symptoms. Overall, agreement between depression diagnoses by GPs and the GDS assessments was weak, regardless of the BMI. These findings suggest that diagnostic agreement could be improved. The standardized evaluation of depression diagnoses in primary care could provide a basis upon which the current clinical diagnostic routine could be enhanced. For instance, improving diagnostic precision for depression in multimorbid older patients in primary care might be achieved through a combination of standardized screening tools (such as the GDS), targeted training, and structured screening of high-risk groups. Trained medical assistants or nurses could, in a first step, identify high-risk patients using a checklist with risk factors and administer a depression screening tool (like the GDS) prior to GP consultation. GPs could then, in a second step, leverage this information to detect (sub)clinical depression, guiding decisions on observation or further diagnostic evaluation. In a third step, identified depression cases could be managed through integrated care pathways involving case managers and mental health specialists. These could, for example, support home-based therapies or interventions to improve problem-solving skills or social and physical activity [61]. A rise in False Positive misidentifications and overtreatment of the respective groups (at the cost of under-provision for other patient groups) could be counteracted through re-evaluation of depression cases over an extended time rather than one-time depression assessments [62]. Before implementation, such approaches would need to be evaluated through feasibility and intervention studies. Collaborative care models, involving mental health specialists, also support the earlier and more effective identification of depression [63]. Finally, allowing more consultation time and building stronger patient–doctor relationships are essential for recognizing depressive symptoms more reliably.

## Figures and Tables

**Figure 1 nutrients-17-01394-f001:**
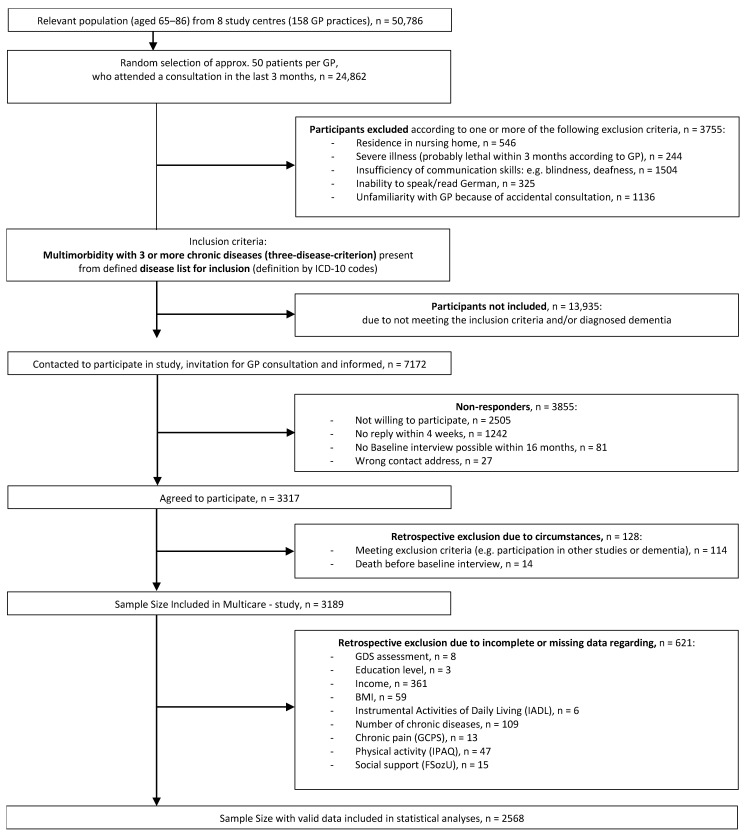
Sample flowchart; total sample of the MultiCare baseline survey.

**Figure 2 nutrients-17-01394-f002:**
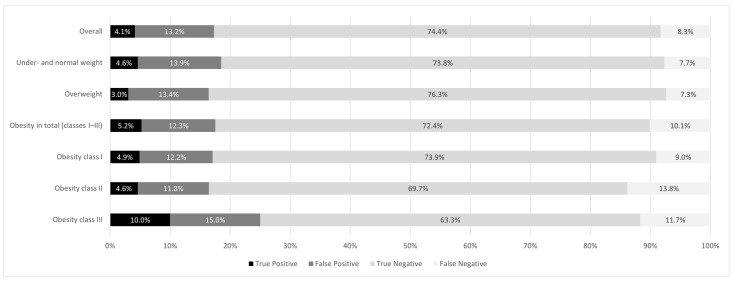
Agreement between GP diagnosis and GDS assessment: percentage distribution of True/False Positives and Negatives.

**Figure 3 nutrients-17-01394-f003:**
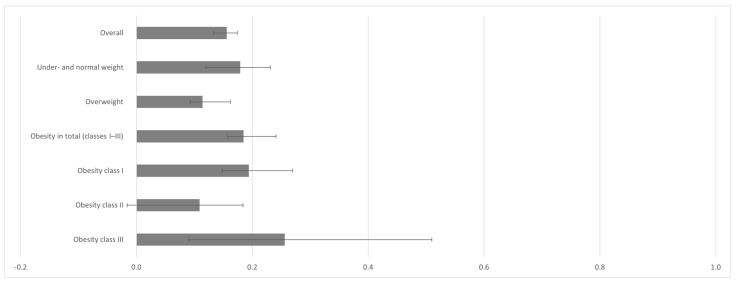
Agreement between GP diagnosis and GDS assessment: Cohen’s kappa with 95% CI as error bars.

**Table 1 nutrients-17-01394-t001:** Sociodemographic and clinical characteristics.

	Overall (*n*, (%))		Women (*n*, (%))		Men (*n*, (%))		*p*-Value
*n* (%)	2568	(100)	1498	(58.3)	1070	(41.7)	
**Age**							0.169 ^a^
<75 years old	1505	(58.6)	861	(57.5)	644	(60.2)	
≥75 years old	1063	(41.4)	637	(42.5)	426	(39.8)	
**Education**							**<0.001 ^a^**
Low	1579	(61.5)	966	(64.5)	613	(57.3)	
Middle	694	(27.0)	443	(29.6)	251	(23.5)	
High	295	(11.5)	89	(5.9)	206	(19.3)	
**Marital status**							**<0.001 ^a^**
Married	1506	(58.6)	664	(44.3)	842	(78.7)	
Single	156	(6.1)	110	(7.3)	46	(4.3)	
Divorced	209	(8.1)	154	(10.3)	55	(5.1)	
Widowed	697	(27.1)	570	(38.1)	127	(11.9)	
**Income**							**<0.001 ^b^**
Mean (s.d.)	1.4	(0.7)	1.3	(0.6)	1.5	(0.8)	
**BMI**							**<0.001 ^a,d^**
Under- and normal weight	627	(24.4)	393	(26.2)	234	(21.9)	
Overweight	1120	(43.6)	579	(38.7)	541	(50.6)	
Obesity in total (I–III)	821	(32.0)	526	(35.1)	295	(27.6)	
Obesity I	609	(23.7)	379	(25.3)	230	(21.5)	
Obesity II	152	(5.9)	101	(6.7)	51	(4.8)	
Obesity III	60	(2.3)	46	(3.1)	14	(1.3)	
**IADL**							0.901 ^a^
Limited	628	(24.5)	365	(24.4)	263	(24.6)	
Not limited	1940	(75.6)	1133	(75.6)	807	(75.4)	
**GCPS pain presence**							**<0.001 ^a^**
Yes	2019	(78.6)	1258	(84.0)	761	(71.1)	
No	549	(21.4)	240	(16.0)	309	(28.9)	
**GCPS pain intensity (0–100)**							**<0.001 ^b^**
Mean (s.d.)	34.0	(25.1)	38.2	(24.8)	28.2	(24.2)	
**Number of chronic diseases**							**<0.001 ^b^**
Mean (s.d.)	7.4	(3.1)	7.7	(3.1)	7.1	(3.2)	
**IPAQ**							**<0.001 ^a^**
Low	805	(31.4)	524	(35.0)	281	(26.3)	
Moderate	1121	(43.7)	647	(43.2)	474	(44.3)	
High	642	(25.0)	327	(21.8)	315	(29.4)	
**Social support**							0.507 ^b^
Mean (s.d.)	4.1	(0.7)	4.1	(0.7)	4.1	(0.7)	
**GDS**							**<0.001 ^c^**
Mean (s.d.)	2.5	(2.6)	2.7	(2.7)	2.3	(2.5)	

NOTE: *n* (%) represents the sample size (*n*) and the percentage (%) of participants/observations in each category. Variables are shown in bold, while their categories are presented in regular font. ^a^ Comparison between women and men based on the chi-squared test, ^b^ comparison between women and men based on the *t*-test, ^c^ comparison between women and men based on the Wilcoxon rank-sum (Mann–Whitney) test, ^d^ comparison between the groups: under- and normal weight, overweight, obesity I, obesity II, and obesity III. Bold formatting of *p*-values is used to indicate statistical significance.

**Table 2 nutrients-17-01394-t002:** Gender- and age-specific depression prevalence according to GP and the GDS questionnaire, distributed by BMI class.

	Overall			Under- and Normal Weight			Overweight			Obesity in Total (Classes I–III)			Obesity Class I			Obesity Class II			Obesity Class III		
	*n* (%)		*p*	*n* (%)		*p*	*n* (%)		*p*	*n* (%)		*p*	*n* (%)		*p*	*n* (%)		*p*	*n* (%)		*p*
**Depression prevalence according to GP**																					
**Total**	444	(17.3)		116	(18.5)		184	(16.4)		144	(17.5)		104	(17.1)		25	(16.5)		15	(25.0)	
**Gender**																					
Female	331	(22.1)	**<0.001**	85	(21.6)	**0.009**	126	(21.8)	**<0.001**	120	(22.8)	**<0.001**	86	(22.7)	**<0.001**	21	(20.8)	0.062 ^a^	13	(28.3)	0.483 ^a^
Male	113	(10.6)		31	(13.3)		58	(10.7)		24	(8.1)		18	(7.8)		4	(7.8)		2	(14.3)	
**Age**																					
<75 years old	274	(18.2)	0.144	78	(21.9)	**0.012**	108	(17.1)	0.498	88	(17.0)	0.611	63	(16.6)	0.674	13	(13.4)	0.182	12	(30.0)	0.343 ^a^
≥75 years old	170	(16.0)		38	(14.0)		76	(15.6)		56	(18.4)		41	(17.9)		12	(21.8)		3	(15.0)	
**Depression prevalence according to GDS questionnaire**																					
**Total**	319	(12.4)		77	(12.3)		116	(10.4)		126	(15.4)		85	(14.0)		28	(18.4)		13	(21.7)	
**Gender**																					
Female	216	(14.4)	**<0.001**	56	(14.3)	0.052	74	(12.8)	**0.006**	86	(16.4)	0.288	59	(15.6)	0.142	18	(17.8)	0.790	9	(19.6)	0.478 ^a^
Male	103	(9.6)		21	(9.0)		42	(7.8)		40	(13.6)		26	(11.3)		10	(19.6)		4	(28.6)	
**Age**																					
<75 years old	170	(11.3)	**0.040**	44	(12.4)	0.945	57	(9.0)	0.095	69	(13.4)	**0.039**	44	(11.6)	**0.030**	13	(13.4)	**0.036**	12	(30.0)	**0.043 ^a^**
≥75 years old	149	(14.0)		33	(12.2)		59	(12.1)		57	(18.8)		41	(17.9)		15	(27.3)		1	(5.0)	

NOTE: *n* represents number of cases. (%) Percentage share is related to the corresponding intersection in each case. Variables are shown in bold, while their categories are presented in regular font. All *p*-values were calculated using the chi-squared test, except for the values labelld with ^a^ calculated using Fisher’s exact test. Bold formatting of *p*-values is used to indicate statistical significance.

**Table 3 nutrients-17-01394-t003:** Results of cross-sectional analyses for the association of depression.

	True Positive Pseudo-R^2^: 0.198			True Negative Pseudo-R^2^: 0.110			False Positive Pseudo-R^2^: 0.035			False Negative Pseudo-R^2^: 0.186		
	OR	(95% CI)	*p* -Value	OR	(95% CI)	*p* -Value	OR	(95% CI)	*p* -Value	OR	(95% CI)	*p* -Value
**Gender** (Ref.: men)												
Women	1.83	(1.05–3.18)	**0.033**	0.58	(0.47–0.73)	**<0.001**	2.01	(1.51–2.68)	**<0.001**	0.96	(0.68–1.36)	0.824
**Age** (Ref.: <75 years old)												
≥75 years old	0.66	(0.43–1.01)	0.057	1.34	(1.09–1.65)	**0.005**	0.79	(0.61–1.02)	0.070	0.99	(0.71–1.38)	0.952
**Marital status** (Ref.: married)		Wald/F: 13.75	**0.003**		Wald/F: 8.68	**0.034**		Wald/F: 2.30	0.513		Wald/F: 0.47	0.925
Single	0.91	(0.35–2.41)	0.855	0.85	(0.57–1.28)	0.437	1.15	(0.70–1.89)	0.587	1.20	(0.66–2.18)	0.553
Divorced	1.57	(0.76–3.27)	0.227	0.67	(0.47–0.95)	**0.024**	1.37	(0.91–2.07)	0.135	1.14	(0.64–2.01)	0.656
Widowed	2.43	(1.47–4.01)	**0.001**	0.75	(0.59–0.94)	**0.015**	1.07	(0.79–1.44)	0.665	1.03	(0.70–1.51)	0.880
**Education** (Ref.: low)		Wald/F: 7.96	**0.019**		Wald/F: 0.72	0.698		Wald/F: 5.77	0.056		Wald/F: 2.98	0.226
Middle	1.05	(0.64–1.72)	0.860	1.06	(0.85–1.32)	0.630	0.78	(0.59–1.03)	0.077	1.31	(0.93–1.86)	0.127
High	2.48	(1.30–4.72)	**0.006**	0.91	(0.65–1.27)	0.571	0.63	(0.40–1.00)	0.050	1.38	(0.79–2.41)	0.253
**Income**	0.90	(0.66–1.22)	0.483	1.38	(1.16–1.64)	**<0.001**	0.96	(0.80–1.16)	0.678	0.47	(0.33–0.66)	**<0.001**
**BMI** (Ref.: ≤ 24.9 kg/m^2^)		Wald/F: 3.11	0.539		Wald/F: 6.24	0.182		Wald/F: 3.25	0.517		Wald/F: 6.83	0.145
Overweight	0.63	(0.36–1.08)	0.092	1.14	(0.89–1.45)	0.312	1.04	(0.78–1.40)	0.769	0.84	(0.56–1.27)	0.413
Obesity I	0.76	(0.42–1.37)	0.366	1.35	(1.02–1.78)	**0.036**	0.82	(0.58–1.15)	0.252	0.75	(0.47–1.19)	0.217
Obesity II	0.63	(0.25–1.56)	0.314	1.21	(0.79–1.84)	0.377	0.77	(0.45–1.33)	0.353	1.08	(0.57–2.04)	0.822
Obesity III	0.65	(0.20–2.07)	0.461	1.79	(0.94–3.41)	0.076	0.93	(0.44–1.99)	0.859	0.33	(0.13–0.84)	**0.020**
**IADL** (Ref.: not limited)												
Limited	3.14	(1.98–5.00)	**<0.001**	0.61	(0.49–0.76)	**<0.001**	0.82	(0.61–1.11)	0.203	2.14	(1.54–2.99)	**<0.001**
**GCPS pain presence** (Ref.: no)												
Yes	1.07	(0.44–2.57)	0.884	0.97	(0.68–1.37)	0.842	1.41	(0.93–2.13)	0.106	0.73	(0.40–1.33)	0.301
**GCPS pain intensity**	1.01	(0.99–1.02)	0.281	0.99	(0.98–1.00)	**<0.001**	1.00	(1.00–1.01)	0.449	1.02	(1.01–1.02)	**<0.001**
**Number of chronic diseases**	1.05	(0.98–1.12)	0.161	0.95	(0.91–0.98)	**0.001**	0.99	(0.95–1.03)	0.509	1.13	(1.07–1.18)	**<0.001**
**IPAQ** (Ref.: low)		Wald/F: 1.63	0.442		Wald/F: 13.18	**0.001**		Wald/F: 1.10	0.576		Wald/F: 13.31	**0.001**
Moderate	0.89	(0.54–1.46)	0.644	1.24	(1.00–1.55)	0.055	1.01	(0.76–1.34)	0.969	0.71	(0.50–1.01)	0.059
High	0.62	(0.30–1.29)	0.202	1.68	(1.27–2.22)	**<0.001**	0.86	(0.61–1.21)	0.395	0.37	(0.22–0.64)	**<0.001**
**Social support**	0.36	(0.28–0.46)	**<0.001**	1.74	(1.51–2.00)	**<0.001**	1.16	(0.97–1.38)	0.102	0.49	(0.40–0.60)	**<0.001**

Notes: Variables are shown in bold, while their categories are presented in regular font. OR = odds ratio; CI = confidence interval; Ref. = reference category. Bold formatting of *p*-values is used to indicate statistical significance.

## Data Availability

The data that support the findings of this study are not publicly available due to privacy and ethical restrictions. The data are, however, available from the authors upon reasonable request and with permission of Martin Scherer (principal investigator).

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
