# Peer review of "Overweight, Obesity, and Depression in Multimorbid Older Adults: Prevalence, Diagnostic Agreement, and Associated Factors in Primary Care—Results from a Multicenter Observational Study"

_nutrients, 2025, doi:10.3390/nu17081394_

Round 1

Reviewer 1 Report

Comments and Suggestions for Authors

This is a very interesting manuscript characterizing the discrepancy between the results of GDS and GP diagnosis of depression based on a multicenter observational study.

It definitely must be stated that GDS is a screening tools, not a diagnostic one. The text needs critical revision as  there are statements which suggest its diagnostic properties e.g. (line 435-437) “We also investigated whether there are differences in identifying depression when comparing the depression diagnosis made and coded by a GP with that of the GDS, a scientifically well- established instrument.”

My main concerns are related to the methodology:

  1. The sample size is rather large, but was the minimal sample size calculated? It needs to be clarified in the text.
  2. How about sensitivity and specificity of GDS vs. depression diagnosis of GP’s? It would be interesting to see at least the ROC curve.

Author Response

We would like to sincerely thank the reviewer for the thoughtful and constructive feedback, which helped us to improve the quality and clarity of our manuscript.

Our detailed response to the reviewer’s comments is provided in a separate file.

Reviewer 2 Report

Comments and Suggestions for Authors

The paper “Overweight, Obesity, and Depression in Multimorbid Older Adults: Prevalence, Diagnostic Agreement, and Associated Factors in Primary Care – Results from a Multicenter Observational Study” investigates the prevalence of depression across different Body Mass Index (BMI) classes and includes age and gender differences in multimorbid older patients. Further the agreement between GP depression diagnoses and the Geriatric Depression Scale (GDS) is studied and patient-specific factors that may affect the agreement are explored. Here are some specific comments.

Comments:

Q1. The analysis was based only on baseline cross-sectional data, making it difficult to establish a causal relationship between obesity and depression.

Q2. Although the cut-off value of ≥6 points used in this article is a common standard, while the basis for selecting this cut-off value and the possible impacts of different cut-off values on the study results should be discussed.

Q3. The study analyzed the impacts of multiple patient-specific factors on depression diagnosis, are there any other unmeasured confounding factors?

Q4. The study concluded that there is a need to improve depression diagnostic practice in primary care, but there is a lack of detailed recommendations on how to specifically implement standardized screening tools, training, and collaborative care models.

Q5. It is suggested that the author more explicitly expound the innovative aspects of the research in the abstract, introduction and conclusion sections to enhance the influence of the research.

Q6. The overall language expression of the article is relatively smooth, but some sentences are overly long and complex, which affects the efficiency of information transmission.

Q7. There seems to be something wrong with the formatting of the numbers in Table 2. Is the first line indented?

Author Response

(The authors gave the same response as above.)

Reviewer 3 Report

Comments and Suggestions for Authors

Dear Authors,

This study was conducted to the overweight, obesity, and depression in multimorbid older adults. Even though this topic between obesity and depression was already well-known in field of medicine and public health section, this study was focused on prevalence, diagnostic agreement, and associated factors in primary care about results from a multicenter observational study which is really interesting study.

Abstract

Line 40: ‘different Body Mass Index (BMI)’ to ‘different body mass index (BMI)’.

Line 46: ‘IC’ to ‘International Classification of Diseases’.

Line 59: ‘OR’ to ‘odd ratio (OR)’; Abbreviations should be defined in the first instance.

Result section: please add 95% confidence interval in each results of OR, respectively.

Sort alphabetically in Key-words.

Line 68: ‘Body Mass Index (BMI); Geriatric Depression Scale (GDS);’ to ‘Body mass index; Geriatric Depression Scale;’

Introduction: Well-written

Line 80: ‘BMI’ to ‘body mass index (BMI)’; Abbreviations should be defined in the first instance.

Method: Well-written

Line 147: The baseline survey took place between July 2008 and October 2009. I think it was too old data. Therefore, please add this limitation in Limitation section.

Line 167: ‘(ICD-10 codes) catalogue’ to ‘(International Classification of Diseases [ICD]-10 codes) catalogue; Abbreviations should be defined in the first instance. Please check same issue in whole manuscript.

Line 204: ‘body mass index (BMI)’ to ‘BMI’. Please check the whole manuscript where this principle was not followed such as term BMI, CP, and GDS etc. For example, Line 208, you have to change ‘Geriatric Depression Scale (GDS)’ to ‘GDS’ because of you already define abbreviations, previously.

Line 209: ‘Instrumental Activities of Daily Living (IADL)’.

Line 234: ‘Instrumental Activities of Daily Living (IADL)’ to ‘IADL’; You already defined it in Line 209. Please check whole manuscript.

Results: Well-written

In Table 2, I hope you align the lines well so that we can understand the Table well.

Discussion: Well-written

You should add more applications from this study.

Furthermore, checking by the iThenticate system, the plagiarism rate was 20% (quotes included and bibliography excluded). If it is possible, please reduce the plagiarism rate under 15%.

Author Response

(The authors gave the same response as above.)
